# External Esophageal Stenting Technique in Palliation for Tracheal Agenesis in a Case of Esophageal Lung: A Lesson Learned from the Experience for Tracheomalacia

**DOI:** 10.3390/children10121907

**Published:** 2023-12-10

**Authors:** Taichi Hirotani, Ryo Tamura, Makoto Ando, Hideaki Okajima

**Affiliations:** 1Department of Pediatric Surgery, Kanazawa Medical University, 1-1 Daigaku, Uchinada, Kahoku 920-0293, Ishikawa, Japan; 2Advanced Pediatric Surgical Center, Kanazawa Medical University, 1-1 Daigaku, Uchinada, Kahoku 920-0293, Ishikawa, Japan; shannon@kanazawa-med.ac.jp

**Keywords:** tracheal agenesis, esophageal lung, external stenting, tracheomalacia, dying spell

## Abstract

Tracheal agenesis (TA) is a rare congenital anomaly with an incidence of 1 per 50,000 newborns. It appears at birth with severe respiratory distress, cyanosis, and inaudible crying. Prompt esophageal intubation and long-term management of the esophageal airway are essential to overcome this catastrophic condition. In the long-term management, external stenting of the esophageal airway has been reported as promising to support the fragile esophageal wall; this technique was taken from the surgery for tracheomalacia. We experienced a case of an infant with tracheal agenesis whose respiratory status was stabilized after external esophageal stenting. The stenting was performed based on a lesson learned in the extensive experience in the surgical treatment for tracheomalacia, and the surgical techniques for successful stenting are herein described.

## 1. Introduction

Tracheal agenesis (TA) is a congenital anomaly in which the whole or partial segment of the trachea is absent [1]. The incidence is estimated at 1 in 50,000 live births [2]. Floyd et al. classified tracheal agenesis into three categories according to the type of fistulous connection with the esophagus [3]. Although TA is associated with premature delivery and polyhydramnios, prenatal diagnosis is difficult. Vertebral defects, anal atresia, cardiac defects, tracheoesophageal fistula, radial or renal anomalies, and limb abnormalities (VACTERL) are known associated anomalies. Most neonates with this condition die within hours of birth owing to difficulties in diagnosis and treatment, in particular, due to the failure of the securement of the airway derived from the absence of the trachea. Mortality is high: 85% of cases die within 2 days [2]. Initial prompt esophageal intubation and the long-term management of the “esophageal airway” are a crucial part of the management. However, in most cases, the management is challenged by the collapsing of the fragile esophagus wall and severe tracheomalacia-like symptoms.

An external stenting technique that provides artificial cartilage to the esophagus to support the floppy esophageal wall has been reported [4,5,6,7]. The technique was initially invented for severe tracheomalacia, and one of the authors (MA) of this article has been extensively involved in the surgical management of tracheomalacia with this technique [8,9]. Although ringed polytetrafluoroethylene (PTFE) has been already used for tracheal external stenting in patients with tracheomalacia, compressive injury on the adjacent structure is possible and may become fatal [9]. Several surgical efforts are necessary to prevent injury of important structures such as large vessels caused by a ringed PTFE graft.

Esophageal lung is an extremely rare type of bronchopulmonary foregut malformation where a main stem bronchus is abnormally connected to the esophagus instead of the trachea. This anomaly may be described by some as being a part of the spectrum of TA [10].

In this case report, we present a case of TA (esophageal lung) in which external esophageal stenting was effective in stabilizing the respiratory status. The external esophageal stenting technique has been already reported in several articles, and this paper presents an additional case to corroborate the effectiveness of the stenting and the important surgical techniques for the success of the procedure based on a lesson learned from the experience for tracheomalacia.

## 2. Case Description

A male infant with a history of polyhydramnios was born at 36 weeks gestational age with a birth weight of 2822 g. The biological parents and the rest of the family had no other significant medical issues including congenital anomalies and intrauterine or perinatal deaths. Respiratory distress developed immediately after birth, and therefore, endotracheal intubation was attempted. The larynx was normally visible, but an endotracheal tube could not go further beyond the subglottic space. TA was suspected, and deliberate esophageal intubation finally succeeded in the resuscitation of the patient. A computed tomography (CT) scan confirmed Floyd type III TA and esophageal lung (Figure 1A). Tetralogy of Fallot (TOF) and anorectal malformation were also diagnosed. An emergency surgery to make the esophagus act as the airway was performed 12 h after the birth. The ligation of the esophagus distal to the origin of the trachea and the resection of the esophageal stenosis just proximal to the bifurcation was carried out through the extrapleural approach. Gastrostomy and colostomy were created simultaneously. To prevent aspiration pneumonia due to the drooling of saliva into the airway via the esophagus, the esophagus was divided at its midpoint, and proximal esophagostomy for saliva drainage and distal esophagostomy for the airway entrance were performed on day 45 of life. His respiratory status was improved after the surgery, but he still suffered from multiple episodes of tracheomalacia-like symptoms due to the floppy esophageal wall, including dying spell episodes, despite the use of an internal stent with a long tracheotomy cannula. This was thought to be due to the collapse of the esophagus distal to the tip of the tracheotomy cannula (Figure 1B,C).

To reinforce the floppy esophageal wall and improve tracheomalacia-like symptoms, external stenting of the esophagus was planned at 7 months of life. The surgery was performed in a supine position with a median sternotomy. Cardiopulmonary bypass was established with an outflow cannula in the aortic arch and inflow cannulas in the superior and inferior vena cava. The right pulmonary artery ran just in front of the esophagus and was mobilized to expose the anterior aspect of the esophagus. A ringed PTFE (16 mm in size) prosthesis was longitudinally cut in half and then adjusted to the length corresponding to that of the collapsed esophageal airway segment determined with intraoperative bronchoscopy. The length of the PTFE graft involved five plastic rings. The edge of the prosthesis was covered with a 0.1 mm Gore-Tex PTFE sheet to prevent contact injury and perforation of surrounding organs (Figure 2A). Then, it was placed around the anterior aspect of the esophagus. Using radial traction sutures, the prosthesis served as an external stent to structurally support the esophagus (Figure 2B,C). Guided by intraoperative bronchoscopy, re-expansion of the collapsed segments was achieved by gentle traction and tying of the 5-0 polypropylene sutures.

After the operation, the re-expansion of the collapsed segment was confirmed in repeat bronchoscopies (Figure 2D). The tip of the endotracheal tube was maintained within the esophagus and stabilized with the external stent by using a custom-ordered long tracheostomy tube. The respiratory status of the patient became stable and the ventilatory support could be weaned within a week after the surgery. At 12 months of age, he still needed ventilator support; however, he was free from dying spell episodes and enjoyed off-ventilator time on multiple occasions a day. His neurological milestones were normal for his age. Enteral nutrition was fully given via gastrostomy, whereas he ate various foods appropriate for his age orally with the well-working proximal esophagostomy. We thought it was possible to perform surgical repair for TOF, and we performed ventricular septal defect (VSD) closure and enlargement of the right ventricular outflow tract at 377 days of age. Postoperatively, pulmonary hypertension worsened, and he underwent removal of the VSD patch and pulmonary artery banding on the 7th day after surgery. But the pulmonary hypertension did not improve, and he died at 386 days of age.

## 3. Discussion

Tracheal agenesis is a rare condition, with 118 English reports of 175 cases since its discovery in 1900 [11]. Most of the patients with this condition died within a few hours after the birth because of respiratory distress and unsuccessful resuscitation, and only ten patients have been reported as surviving more than 12 months to date [4,5,7,11,12,13,14,15].

Our patient had Floyd’s type III TA. Floyd et al. classified TA based on the type of fistulous connection with the esophagus into three types [3]. In type I, the stump of the distal trachea attaches to the anterior esophageal wall. Type II involves no trachea, but a carina is present with a fistula to the esophagus. In type III, two mainstem bronchi arise individually from the esophagus. The relative incidence of these three types is 19%, 51%, and 30% [11]. Floyd’s type III TA may be described as esophageal lung [10].

From a viewpoint of the management of this near-lethal condition, initial securement of the airway plays a crucial role, and prompt esophageal intubation would be life-saving [2]. Emergent surgery after the temporary securement of the airway is mandatory to divide the esophagus just distal to the origin of the trachea to use the esophagus as a new airway tract. The resection of the stenotic part in the esophagus just proximal to the divergence of the trachea is also required if the stenotic part hampers stable ventilation [14]. Further esophageal diversion at the cervical esophagus to prevent aspiration pneumonia due to the draining of saliva into the esophageal airway is carried out at the same time or later [15]. These procedures are performed in an acute or semi-acute setting to stabilize the respiratory status of the patient, whereas the management of tracheomalacia-like symptoms would be a long-term issue afterward since the esophageal wall lacks cartilage which keeps the lumen patent. The symptoms would be so severe that the patient may suffer a “dying spell” in the worst scenario [16]. External stenting for tracheomalacia was first described in 1997 and then included in the treatment of TA in 2008 [4,17]. In this procedure, artificial material such as a PTFE vascular graft is sutured to the esophageal wall. The technique has been described in a limited number of articles but has gradually been acknowledged [4,5,6,7].

Five TA patients underwent external stenting at 2 to 3 months of age in three cases and 7 to 8 months of age in two cases (Table 1). Two Floyd I patients and one Floyd II patient (case 4) required resection of a tracheoesophageal fistula and reanastomosis with external stenting. In a Floyd II patient (case 3), the stenotic tracheoesophageal fistula was widened using a pericardial patch with external esophageal stenting. In case 1 and case 4, full-circle PTFE grafts were placed around the esophagus, whereas a semicircle PTFE graft was placed on the anterior aspect of the esophagus in case 2 and case 5. In case 3, a 3/4-circle PTFE graft was placed around the esophagus other than the part covered by a pericardial patch. A semicircle graft for the anterior wall allows airway growth, maintains adequate blood supply to the esophagus, and enables us to use a smaller prosthesis. This is important for avoiding contact injury and perforation of surrounding organs such as large vessels, especially in pediatric patients whose organs are in the vicinity. A semicircle graft around the anterior wall may be sufficient for the re-expansion of the collapsed esophagus because the posterior wall is supported by vertebrae. 

In the current case, the stenting was performed by one of the authors (MA) who has reported his 98 cases of experience of external stenting for tracheomalacia [9]. Based on his substantial experience, several surgical techniques were introduced for successful stenting. Firstly, the approach for the exteriorization of the esophagus was via median sternotomy. For pediatric surgeons, posterolateral thoracotomy is a familiar approach for handling the thoracic esophagus; however, in this case, extensive adhesions from the previous thoracotomy were expected, and disturbance of blood supply after the adhesiolysis around the esophagus was concerning. In particular, the lack of blood supply after the adhesiolysis was considered to be detrimental since the esophagus was divided multiple times in the previous surgeries, and it was difficult to anticipate normal vascular anatomy supplying the organ. Hence, the anterior aspect of the esophagus which was intact from the previous surgeries was approached directly. Secondly, the anterior aspect of the esophagus where the external stent would be applied was dissected meticulously from the surrounding vessels like the pulmonary artery guided by intraoperative bronchoscopy. Intraoperative bronchoscopy was indispensable to determine the length of the esophagus which should be stented and directly confirm the patency of the esophagus after the application of the sutures between the esophagus and the stent. The last tip was the covering of the edge of the PTFE stent with a 0.1 mm PTFE Gore-Tex sheet to prevent direct contact injury of the surrounding vessels and the esophagus by the stent.

After external esophageal stenting, it is important to maintain the tip of the endotracheal tube within the esophagus stabilized with the graft. A custom-ordered long tracheostomy tube would be one of the solutions if a ready-made tube could not fit well into the newly formed trachea. Careful follow-up bronchoscopy is necessary to prevent luminal obstruction due to the ulceration of the esophageal mucosa or the formation of granulation tissue. Management of TA requires a multidisciplinary approach. TA patients have a stormy postoperative course including multiple surgeries and a prolonged postoperative course. An experienced anesthetist, a specialized pediatric bronchoscopist, a dedicated surgical team, and a trained intensive care unit team are essential in the management of this lethal and precipitous condition.

Regarding the long-term prognosis of TA, mechanical respiratory support and oxygen administration were not necessary in a few long-term survivors [15], whereas some patients died because of airway troubles such as accidental extubation [11], airway obstruction due to esophageal ulceration and granulation at the distal end of the airway tube [12], and a total collapse of the reconstructed trachea [6]. In TA patients achieving stabilization of the respiratory status, esophageal/alimentary reconstruction using a gastric tube, the jejunum, or the patient’s own esophagus was performed to enable oral feeding. More surprisingly, one of these patients gained esophageal speech technique despite the congenital absence of a trachea [15]. These managements are necessary to improve the quality of life of long-term survivors with TA.

Tracheal replacement therapy, other than using the esophagus, could be proposed as a solution for patients with TA. Autologous tissue transplantation [18], tissue flaps [19], tracheal allotransplantation [20], bioprosthetic reconstruction, and tissue-engineered medicine [21] are potential approaches. However, these are currently inadequate. The absence of a ciliated epithelial lining to provide mucociliary clearance remains a limiting aspect of using autologous tissue, which has no ciliated epithelium. Allotransplantation also involves long-term immunosuppression with concomitant risks. The ideal repair would be replacement using a bioengineered tracheal graft, but it has not yielded a reliable solution.

## 4. Conclusions

External stenting of the esophagus would be effective in relieving tracheomalacia-like symptoms in a patient with an esophageal airway. Several surgical tips including the surgical approach, intraoperative bronchoscopy, and the covering of the edge of ringed PTFE graft were learned from the experience of external stenting for tracheomalacia, and they worked as well in the current case of tracheal agenesis with a floppy esophageal airway. It would be worth considering the employment of these techniques if surgeons encounter this rare and catastrophic condition. As tracheal agenesis is a rare and lethal disease, prospective studies are difficult, but we believe this report will be useful in future analytical studies.

## Figures and Tables

**Figure 1 children-10-01907-f001:**
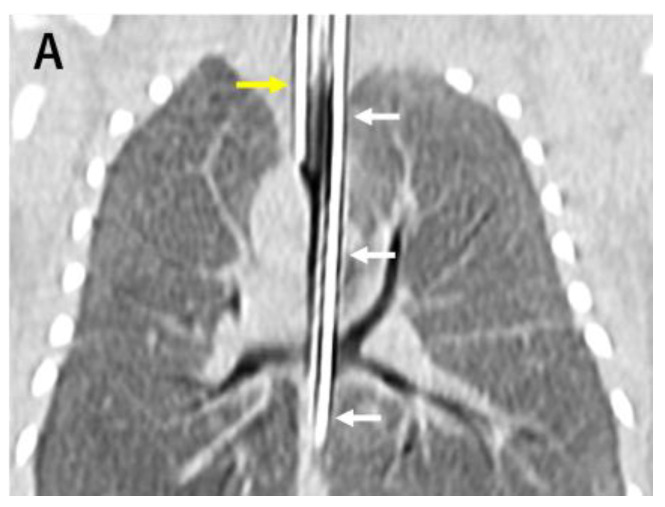
(**A**) CT scan at birth showed esophageal intubation (yellow arrow) and gastric tube (white arrow). (**B**) Bronchoscopy at 1 month of age demonstrated collapsed airway. (**C**) Three-dimensional CT scan at 5 months of age revealed the esophageal airway (arrows). (**D**) Preoperative bronchoscopy at 6 months of age showed that the airway remained collapsed.

**Figure 2 children-10-01907-f002:**
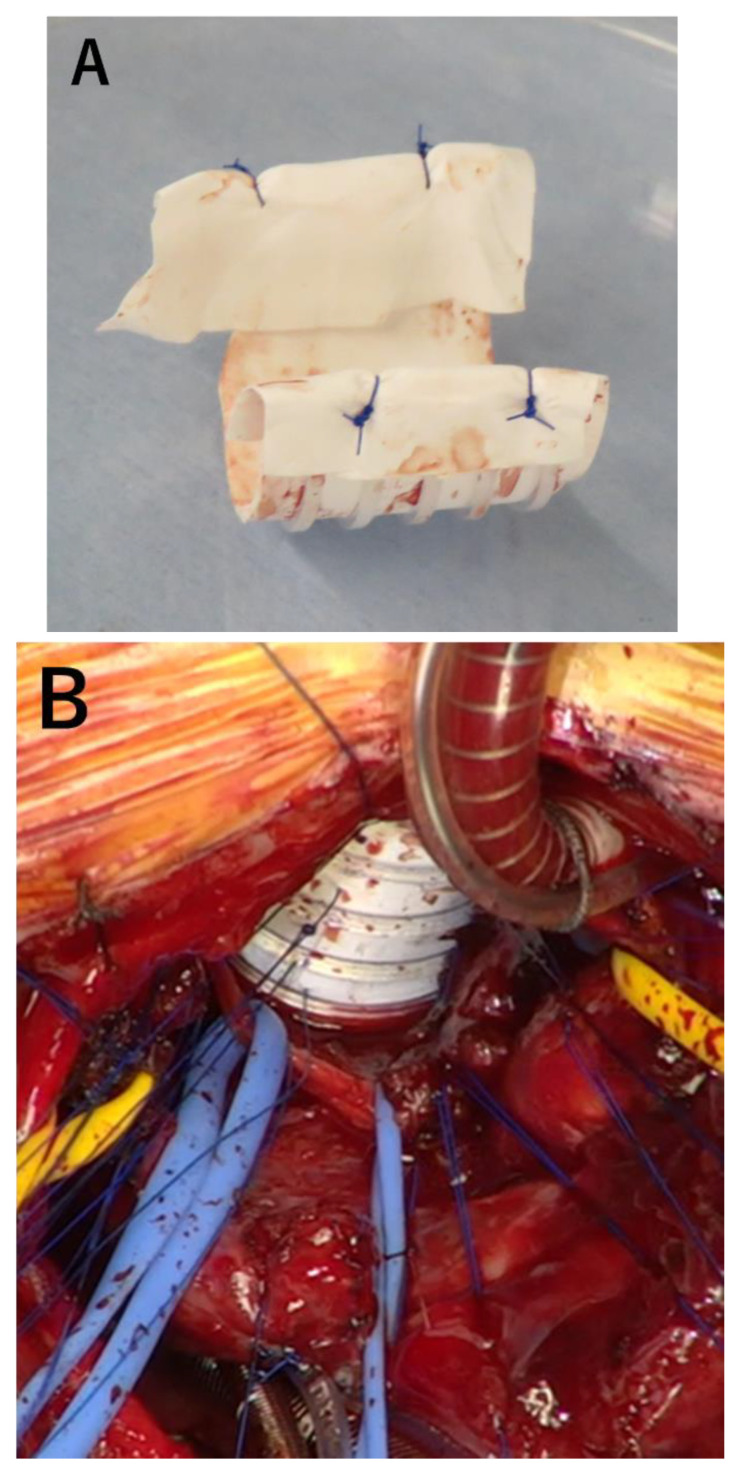
(**A**) The edge of the prosthesis was covered with a 0.1 mm Gore-Tex PTFE sheet. (**B**) The surgical field during the external stenting for the esophagus through median sternotomy. (**C**) On postoperative 3-dimensional CT scan, a ringed polytetrafluoroethylene prosthesis (arrow) was placed around the anterior aspect of the esophagus. The tip of the custom-ordered long tracheostomy tube was maintained within the esophagus and stabilized with the external stent. (**D**) Postoperative bronchoscopy at 8 months of age demonstrated the patent airway. The carina (white arrow) and the right main bronchus (yellow arrow) could be visualized. (**E**) Bronchoscopy at 11 months of age also showed the carina (white arrow) and right main bronchus (yellow arrow).

**Table 1 children-10-01907-t001:** Reported TA cases that underwent external esophageal stenting.

Case	Publication	Floyd’s Type	Age of External Stenting (Months)	Surgical Approach	PTFE Graft Size and Form	Reported Age of Survival (Months)
1	2008 [4]	I	2	Median sternotomy	16 mm, full circle	109
2	2010 [5]	I	8	Right thoracotomy	20 mm, semicircle	98
3	2015 [6]	II	3	Median sternotomy	8 mm, 3/4 circle	11
4	2017 [7]	II	3	Median sternotomy	20 mm, full circle	36
5	2023 *	III	7	Median sternotomy	16 mm, semicircle	12

* Current case.

## Data Availability

The datasets used and/or analyzed during this study are available from the author upon reasonable request.

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
