# Peer review of "External Esophageal Stenting Technique in Palliation for Tracheal Agenesis in a Case of Esophageal Lung: A Lesson Learned from the Experience for Tracheomalacia"

_children, 2023, doi:10.3390/children10121907_

Round 1

Reviewer 1 Report

Comments and Suggestions for Authors

Dear Editors,

Thank you very much for the opportunity to review this article describing an interesting clinical case.

Clinical case publications are often undervalued because they may not be included in accreditation processes. However, they are crucial. Even if these articles do not introduce new information, they describe a novel technique in detail, adding to the existing literature on a procedure for a specific pathology with a low incidence, making it difficult to obtain case series for prospective analytical studies.

The value of these communications is that they contribute to the dissemination of the technique and help gather enough cases to perform subsequent analyses based on literature reviews. Therefore, this article is interesting and necessary.

However, I suggest revising minor aspects to improve the text:

The introduction is brief but sufficient as it summarizes the two topics discussed. However, although the authors acknowledge that their case is only a contribution, they could have provided a better justification for communicating their case. For instance, they could have argued some of the reasons I mentioned in the previous paragraph. It is crucial to communicate such cases since the low incidence of the pathology, coupled with the novelty of the technique, makes it almost impossible to conduct prospective analytical studies.

At the beginning of the description of the clinical case, it would be interesting to know if, in the biological parents, as in the rest of the family, there are other congenital anomalies diagnosed or intrauterine or perinatal deaths that have not been well-documented.

The conclusions offered contain some general aspects of esophageal atresia and the technique. However, the conclusions should focus on the technique employed and what the authors have learned from the clinical case.

As this article is not a study, the value of the conclusions lies in what the authors have gleaned from the clinical case. Therefore, to write the conclusions, I encourage the authors to concentrate on specific aspects of their case that could assist other professionals rather than trying to draw general conclusions that may not be relevant to other cases.

Author Response

Dear reviewer

Thank you very much for reviewing our manuscript and offering valuable advice. We have addressed your comments with point-by-point responses and revised the manuscript accordingly.

Comment 1: The introduction is brief but sufficient as it summarizes the two topics discussed. However, although the authors acknowledge that their case is only a contribution, they could have provided a better justification for communicating their case. For instance, they could have argued some of the reasons I mentioned in the previous paragraph. It is crucial to communicate such cases since the low incidence of the pathology, coupled with the novelty of the technique, makes it almost impossible to conduct prospective analytical studies.

Response: Thank you very much for your excellent suggestion. We agree with you and have incorporated your suggestion in our paper. We have therefore added the following text to the Conclusion section (p.8, line 228-230).

“As tracheal agenesis is a rare and lethal disease, prospective studies are difficult, but we believe this report will be useful in future analytical studies.”

Comment 2: At the beginning of the description of the clinical case, it would be interesting to know if, in the biological parents, as in the rest of the family, there are other congenital anomalies diagnosed or intrauterine or perinatal deaths that have not been well-documented.

Response: The biological parents and rest of family had no other significant medical issues including congenital anomalies, intrauterine or perinatal deaths. We have added the following text to the Case Description section (p.2, line 53-54).

“The biological parents and rest of family had no other significant medical issues including congenital anomalies, intrauterine or perinatal deaths.”

Comment 3: The conclusions offered contain some general aspects of esophageal atresia and the technique. However, the conclusions should focus on the technique employed and what the authors have learned from the clinical case.

As this article is not a study, the value of the conclusions lies in what the authors have gleaned from the clinical case. Therefore, to write the conclusions, I encourage the authors to concentrate on specific aspects of their case that could assist other professionals rather than trying to draw general conclusions that may not be relevant to other cases.

Response: We appreciate your pertinent comment. In the Conclusions section (p8, line 222), we have removed the following sentence which contains general aspects of the disease.

“Tracheal agenesis seems rare but is important as differential diagnosis for respiratory distress in a case of a newborn with inaudible crying and inability to intubate into the trachea in spite of good visualization of the glottis.”

Thank you again for your comments on our paper. We trust that the revised manuscript is suitable for publication.

Reviewer 2 Report

Comments and Suggestions for Authors

I read with great interest this case report on Tracheal agenesis that was managed by using the oesophagus and splinting it externally using a PTFE stent. The topic is of interest, though it has been discussed by way of case reports in the literature. There are several points which need improvement:

- As the authors have mentioned in the Discussion section, the major problems in a tracheal replacement surgery is the lack of mucociliary clearing respiratory epithelium. This is one single factor that decides the final outcome, apart from external scaffolding of the esophagus and its vascularity. My question to the authors is whether they did any biopsies in the replacement, and if no...why not and if yes..what did they find?

- The Methods section and the case reporting needs more clarity, especially the Fig 2C. Did the authors perform a tracheostomy with a customed tube passed into the trachea and the PTFE splint on the outside (as seen by the white arrow). This needs to be clarified.

- And finally, what are the timelines when they did the endoscopic photos that have been submitted. Can they share early- mid range and long term photos to show the evolution of the tracheal replacement.

Author Response

Dear reviewer

Thank you very much for reviewing our manuscript and offering valuable advice. We have addressed your comments with point-by-point responses and revised the manuscript accordingly.

Comment 1: As the authors have mentioned in the Discussion section, the major problems in a tracheal replacement surgery is the lack of mucociliary clearing respiratory epithelium. This is one single factor that decides the final outcome, apart from external scaffolding of the esophagus and its vascularity. My question to the authors is whether they did any biopsies in the replacement, and if no...why not and if yes..what did they find?

Response: Thank you for providing these insights. We did not perform biopsies of the esophagus or other tissues. As you pointed out, the biopsy of the esophagus may provide new insights into the pathology of tracheal agenesis. Regrettably, however, during the surgery performed 12 hours after birth, there was no time to perform a biopsy as it was an emergency surgery. During the surgery at 45 days old, we did not perform a biopsy because we wanted to preserve the esophagus as long as possible for future gastrointestinal reconstruction. 

Comment 2: The Methods section and the case reporting needs more clarity, especially the Fig 2C. Did the authors perform a tracheostomy with a customed tube passed into the trachea and the PTFE splint on the outside (as seen by the white arrow). This needs to be clarified.

Response: Thank you very much for your excellent suggestion. We used custom-ordered long tracheostomy tube to maintain the tip of endotracheal tube within the esophagus stabilized with the PTFE splint on the outside. We have therefore added the following text to the Case Description section (p.5, line 113-115).

“The tip of endotracheal tube was maintained within the esophagus stabilized with the external stent by using the custom-ordered long tracheostomy tube.”

Similarly, we have added the following text in explanatory title of Figure 2C (p.5, line 105-107).

“The tip of custom-ordered long tracheostomy tube was maintained within the esophagus stabilized with the external stent.”

Comment 3: And finally, what are the timelines when they did the endoscopic photos that have been submitted. Can they share early- mid range and long term photos to show the evolution of the tracheal replacement.

Response: Thank you for your suggestions. The endoscopic examinations shown in Figure. 1D and Figure. 2D were performed at 6 and 8 months of age, respectively. We have therefore added text to indicate when the endoscopy was performed in the explanatory title of Figure. 1D (p.3, line 77) and Figure. 2D (p.5, line 107). Further, we have added Figure 1B showing the early photo at 1 month of age and Figure 2E showing the long term photo at 11 months of age. Along with this, we added the following texts in explanatory title of Figure 1B (p.3, line 76-77) 

“(B) Bronchoscopy at 1 month of age demonstrated collapsed airway.”

and Figure 2E (p.5, 109-110).

(E) Bronchoscopy at 11 months of age also showed the carina (white arrow) and right main bronchus (yellow arrow).

And, we have changed the following text in explanatory title of Figure 1 (p.3, line 78-79):

“(C) Preoperative bronchoscopy demonstrated collapsed airway.”

to

“(D) Preoperative bronchoscopy at 6 months of age showed that the airway remained collapsed.”

Thank you again for your comments on our paper. We trust that the revised manuscript is suitable for publication.

Round 2

Reviewer 2 Report

Comments and Suggestions for Authors

Thank you to the Authors for their corrections.

After going through the revised submission, I have the following comments:

-       Without a biopsy, actual trachealisation cannot be confirmed. The tracheostomy and external PTFE splint kept the oesophageal lumen open and acted as a conduit up to the lower airway. The tracheostomy cannula could have stented the oesophagus and perhaps the external splint may not have been even required.

-      The submission must be titled as ‘Stented oesophagus in palliation for tracheal agenesis’
